# Underlying Biochemical and Molecular Mechanisms for Seed Germination

**DOI:** 10.3390/ijms23158502

**Published:** 2022-07-31

**Authors:** Muhammad Awais Farooq, Wei Ma, Shuxing Shen, Aixia Gu

**Affiliations:** State Key Laboratory of North China Crop Improvement and Regulation, Key Laboratory of Vegetable Germplasm Innovation and Utilization of Hebei, Collaborative Innovation Center of Vegetable Industry in Hebei, College of Horticulture, Hebei Agricultural University, Baoding 071000, China; awaisfarooq724@gmail.com (M.A.F.); mawei0720@163.com (W.M.)

**Keywords:** seed germination and dormancy, phytohormones, light, temperature and endosperm decay

## Abstract

With the burgeoning population of the world, the successful germination of seeds to achieve maximum crop production is very important. Seed germination is a precise balance of phytohormones, light, and temperature that induces endosperm decay. Abscisic acid and gibberellins—mainly with auxins, ethylene, and jasmonic and salicylic acid through interdependent molecular pathways—lead to the rupture of the seed testa, after which the radicle protrudes out and the endosperm provides nutrients according to its growing energy demand. The incident light wavelength and low and supra-optimal temperature modulates phytohormone signaling pathways that induce the synthesis of ROS, which results in the maintenance of seed dormancy and germination. In this review, we have summarized in detail the biochemical and molecular processes occurring in the seed that lead to the germination of the seed. Moreover, an accurate explanation in chronological order of how phytohormones inside the seed act in accordance with the temperature and light signals from outside to degenerate the seed testa for the thriving seed’s germination has also been discussed.

## 1. Introduction

The germination of seeds plays a significant role in crop production, and it is an intricate process that occurs due to the precise optimization of endogenous (phytohormones, endosperm decay) and exogenous factors (light and temperature) [1]. The transition from dormancy to germination begins when the dry seed comes in contact with water and ends when the radicle has emerged through all the coats of the developing embryo [2]. This encounter activates the internal metabolic process, involving the careful equilibrium phytohormones [3] in the presence of optimum light and temperature to overcome the seed’s dormancy [4] Nevertheless, a deeper and sequential understanding of the interplay of intrinsic and extrinsic factors for seed germination is a prerequisite for the improvement of seed germination potential in various crops.

Abscisic acid (ABA) and gibberellic acid (GA) play a key role in a number of physiological processes during seed germination [5]. ABA induces dormancy, while GA plays a key role in the release of dormancy and germination. A high ABA:GA ratio maintains dormancy, while dormancy release involves a net shift to increased biosynthesis of GA and ABA degradation resulting in a low ABA:GA ratio. These two hormones may also act in an antagonistic manner in the promotion of testa and endosperm rupture [6].

The degree of seed dormancy is established during seed maturation and governs the behavior of the seed after shedding or even while still attached to the mother plant [7]. The initiation of seed dormancy is coordinated in zygotic tissues by environmental factors that also perform overlapping roles in the control of embryonic identity, storage reserve accumulation, and onset of desiccation tolerance. In addition to the metabolic pathways triggering under the influence of environmental cues, for the seed to rupture, it must go through a series of checks in a sequential manner that leads to testa and endosperm rupture. In many species, seed covering layers impose a physical constraint to radical protrusion, which must be overcome by the growth potential of the embryo [8].

Numerous extrinsic factors can prolong or terminate seed dormancy and promote seed germination and development. Light [9], temperature [10], and soil conditions [11] are major signals that can be perceived by seeds to regulate the timing of germination. The regulatory effect of light on seed germination depends on its spectrum [12]. Blue light activates ABA and delays seed germination, whereas red or far-red light plays a key role in the activation of seed germination via the activation of GA biosynthesis and restricting the production of ABA [13]. Seed germination is dependent on the surrounding temperature, which can delay or expedite the germination process after sowing. Most species germinate in the presence of temperatures between 15–30 °C [14]. Weakening of the endosperm is a prerequisite for the initiation of seed germination and is driven by various internal and external factors. The decay of the endosperm is directly linked to the production of ROS in response to the availability of external environmental signals [15]. This article summarizes the roles of environmental factors (temperature and light), phytohormones, and endosperm decay in seed germination (Figure 1).

## 2. Main Text

### 2.1. Phytohormone Regulation of Dormancy and Germination

Seed germination and dormancy are strictly regulated by hormones, particularly ABA and GA, which have antagonistic effects. ABA biosynthesis takes place mainly due to carotenoid cleavage dioxygenase gene 6 (NCED 6), NCED 9, and other genes, such as ABA-deficient (ABA2) and abscisic aldehyde oxidase 3 (AAO3) during seed development to maintain seed dormancy. ABA-insensitive (ABI) genes, i.e., ABI3, ABI4, and ABI5 are more plentiful in dormant seeds than in seeds with reduced seed dormancy levels. Among ABI genes, ABI3 expressed in the developing seeds also regulates the accumulation of chlorophyll, anthocyanins, and storage proteins together with two other seed-related regulators, FUSCA3 (FUS3) and leafy cotyledon 1 (LEC1). ABA levels in seeds depend on the degradation process. The catabolism of ABA is mediated by ABA8 hydroxylase encoded by P450 (CYP707A) genes and is induced by imbibition and stratification; the concentration of ABA declines and that of GA increases [16]. ABA-dependent pathways are also very important in the regulation of seed germination. ABA signaling must be terminated by a process in which the membrane-associated transcription factor peptidases S1P (Site-1 Protease) and S2P translocate the bZIP17 protein from the endoplasmic reticulum (ER) to the Golgi apparatus and then to the nucleus. There, activated bZIP17 regulates the transcription of downstream negative regulators of ABA signaling [17]. ABA acts through the PYR/PYL/-RCAR-PP2C-SnRKs signaling cascade. The PP2C proteins ABI1 and ABI2 bind to ABA receptors to inhibit signaling [18,19]. Their dominant negative mutants, abi1-1 and abi2-2, show reduced dormancy due to the impaired interaction between the mutated proteins and their receptors. The other PP2C HONSU (HON) protein also influences seed germination by downregulating ABA signaling and upregulating GA signaling [20]. In addition, another PP2C gene, RD05 (reduced dormancy 5), has a positive role in the reduction of seed dormancy [21]. Genetic and bioinformatic analysis showed that RD05 controls seed dormancy by mediating transcription of the PUF family RNA-binding genes APUM9 (Arabidopsis PUMILI09) and APUM11 [21]. However, RD05 appears to function independently of the ABA pathway, and further research is needed to accurately delineate its role in seed germination processes.

GA biosynthesis mainly occurs in the radicle of the embryo, which in turn ensures germination progression. However, high exogenous concentrations of GA can negatively influence the germination process [22]. The activation of GA-responsive genes induces cell-wall-remodeling enzymes, such as endo-β mannase, xyloglucan endotransglycolase, expansin, and β 1,3-mannase. Their activity leads to the weakening of the surrounding embryo layers. GA breaks dormancy by antagonistically suppressing ABA-triggered seed dormancy. This process appears to involve the secretion of hydrolytic enzymes gibberellin 3-oxidase 1 (GA3ox1), GA20ox3, and ENT-kaurene oxidase 1 (KO1) to weaken the seed testa, although detailed and precise information about this mechanism is lacking. GA-deficient mutants, such as ga1 and ga2, show strong dormancy and cannot germinate without external GA application [22]. Mutations in DELLA genes, including RGL2 (RGA-LIKE2) and SPY (SPINDLY), two negative regulators of the GA signaling pathway, can rescue the non-germination phenotype of ga1. DELLAs also maintain the seed in a quiescent state of cell cycle progression by repressing the activities of TCP14 (Teosinte branches 1/cycloidea/proliferating cell factor) and TCP15 [23]. The sleepy1 (SLY1) is an F-box protein which enables 26S-proteasome-mediated degradation of DELLA proteins in the presence of active GA. *sly1* mutants show reduced germination even after the application of exogenous GA [24].

Genes in hormone signaling pathways also play an important role in regulating seed germination. The expression levels of numerous genes are up- and downregulated to mediate seed dormancy and germination (Table 1). The APETALA 2 (AP2) domain containing transcription factor ABI4 plays a significant role in seed dormancy regulation. ABI4 controls many signaling pathways, including responses to ABA, glucose, sucrose, ethylene (ET), and salt stress. ABI4 positively regulates ABA catabolism genes and negatively regulates GA biogenesis genes. The loss of ABI4 function increases the expression of GA biosynthesis genes but decreases the expression of GA inactivation genes; together, these changes lead to decreased primary seed dormancy in the abi4 mutant [25]. Furthermore, ABI4 binds to the promoters of CYP707A1 and CYP707A2, which mediate ABA catabolism, inhibiting their transcription and thereby promoting the accumulation of ABA. However, no direct role of ABI4 in GA metabolism has been identified, and ABI4 may not directly bind to the promoters of GA biosynthesis genes. Instead, it may recruit additional seed-specific transcription factors to repress the transcription of GA metabolism genes [26]. Clearly, the AP2 domain plays an important role in the dual regulation of ABA and GA biosynthesis to optimize seed dormancy and germination.

Auxin is involved in all stages of plant development and in the response to a multitude of environmental cues. Exogenous auxin application triggers seed dormancy under salt stress, indicating its role in seed germination [53]. IAA (indole-3-acetic acid) delays seed germination and inhibits preharvest sprouting in wheat [54]. Seed dormancy and germination are controlled by auxin-related genes. Biochemical studies have shown that when auxin levels are low due to the suppression of auxin-responsive factors 10 (ARF10) and ARF16 by AXR2/3, the expression of AB13 is not activated and seed dormancy is released. Contrarily, when auxin levels are high, ARF10 and ARF16 are released to activate AB13 transcription and seed dormancy is maintained [55]. Increases in the biosynthesis of auxins can be linked to the release of dormancy in monocots. Furthermore, TaAuxin-resistant 1 (TaAXR1), TaUbiquitin-related protein 1 (TaRUB1), and TaARF@ were upregulated in the ripened wheat seeds. TaAXR1 is associated with AUX/IAA-proteasome-mediated degradation, whereas TaRUB1 is related to ubiquitin action. The higher expression of TaAXR1 and TaRUB1 can exert a negative impact on auxin signaling [46].

Ethylene (ET) breaks seed dormancy and enhances seed germination by reducing the effects of ABA. Changes in positive regulators of the ET signaling pathway cause severe dormancy, whereas mutations in the negative ET regulator Ctr1 (Constitutive Triple Response 1) lead to rapid or early seed germination [56]. Brassinosteroids (BRs) have been reported to act in opposition to ABA to improve seed germination, partly through an MFT (MOTHER OF FT and TFL1)-mediated pathway that generates a negative feedback loop to modulate ABA signaling. The ABA response of BR mutants or BR-deficient plants is stronger than that of wild-type seeds, indicating that BR overcomes the inhibitory effect of ABA on germination [57]. Salicylic acid (SA) controls seed germination by inhibiting the expression of GA-induced α-amylase genes under normal growth conditions. Moreover, SA has been found to promote seed germination under salt stress through another signaling pathway that reduces oxidative damage [58]. Cytokinins (CTKs) improve seed germination by reducing the impact of ABA, specifically by the downregulation of AB15 transcription. A recent study has shown that AB15 plays an important role in ABA and CTK signaling at both the mRNA and protein levels [59]. Jasmonic acid (JA) has an antagonistic relationship with ABA: it not only suppresses ABA biosynthesis genes but also inhibits ABA inactivating genes (Figure 2) [60].

### 2.2. Light Controls Seed Germination and Dormancy

Light is indispensable for germination, although the exact functions of light in seed germination require additional study. Light regulates various plant physiological processes, such as seed germination and dormancy, photomorphogenesis, phototropism, and flowering. There are many factors involved in light regulation of seed dormancy and germination (Table 2). At least five kinds of photoreceptors in plants that monitor surrounding light signals have been reported [14]. Blue light (320–500 nm wavelength) is absorbed by photosensory receptors, including the cryptochromes (CRYs), FLAVIN-BINDING KELCH REPEAT F-BOX1/ ZEITLUPE /LOV KELCH PROTEIN2, and phototropins. Numerous genetics studies have indicated that alterations in these photoreceptors can cause changes in seed germination and agronomical traits. Blue light has been identified to play a role in seed germination inhibition. Cryptochrome 1 mediates this inhibition by downregulating CRY1a and CRY1b products in barley through an RNA interference (RNAi) approach that results in reduced blue light inhibition of grain germination, suggesting the specific role of cry1 in promoting seed dormancy in this monocot. This effect is due to the induction of ABA biosynthetic gene 9-CIS-EPOXYCAROTENOID DIOXYGENASE 1 (NCED1) with consequent ABA biosynthesis. Blue light does not induce NCED 1 in germinating CRY1a/CRY1b RNAi seed, whereas ABA-catabolic gene ABA8′OH-1 is upregulated during early phases of germination [61,62]. The blue light receptor cryptochrome circadian regulator 1 (CrY1) mediates the stimulatory effects of blue light on the expression of NCED1, which increases ABA content and inhibits seed germination in dormant barley [61,63]. Blue light has also been reported to inhibit seed germination in Brachypodium distachyon [14]. Previous studies have suggested that blue light represses seed germination by enhancing the transcription of ABA biosynthesis genes and repressing the expression of ABA catabolism genes [60].

Phytochromes are necessary for the light-induced promotion of seed germination. PhyB occupies the most important position. PhyB mediates the red/far-red photo-reversible response (LFR) to induce the early stages of seed germination. In response to long nights and imbibition, phyA mediates the very low fluence response (VLFR) to different light spectra (UV-A-FR) and the R/FR high irradiance response (R/FR-HIR) to accelerate seed germination in the absence of active phyB [14]. PhyA- and phyB-dependent germination induction are spatially separated and occur in the endosperm and embryo, respectively [86]. PhyE is required for germination in the presence of continuous far-red light [87]. PhyE and phyD stimulate germination at very low red/far-red ratios, whereas phyC antagonizes the promotion of germination by light [65].

Basic helix–loop–helix (bHLH) transcription factors from the PHY-INTERACTING FACTORS (PIF) family negatively regulate the phytochrome-mediated light-signaling pathway [84]. The Arabidopsis genome encodes eight PIFs: PIF1, PIF2/PIL1, and PIF3–PIF8. The interaction between the phytochromes and the PIFs depends mainly on short domains located in the amino termini of the PIFs: APB for Pfr phyB binding and APA for Pfr phyA binding. Light-activated Phys modulate the functions of PIFs through different mechanisms. For example, the phyB-PIF interaction lowers the DNA-binding capacity of PIF1, PIF3, and PIF4 [88]. However, there is still much to learn about the regulation of seed germination by light of different wavelengths in order to devise appropriate strategies for individual plant species.

Red or far-red light (600–750 nm) is perceived by the phytochromes [89], and UV-B light (280–320 nm) is perceived by UV RESISTANCE LOCUS8 (UVR8). These photoreceptors mediate light signals to remodel global transcriptional programs by selectively interacting with transcription factors or E3 ubiquitin ligases that regulate the stability of transcription factors [90]. Studies have demonstrated that red and far-red light modulates seed germination through interactions between phys and PIF1, which in turn controls ABA and GA pathways [91]. Phytochromes were first identified in lettuce as regulators of seed germination. Red light can induce seed germination, whereas far-red light has an inhibitory effect [92]. These effects have been confirmed repeatedly in multiple plant species. Both the inactive form (Pr) and the active form (Pfr) of phytochromes are present in plants. Pr is converted into Pfr by the absorption of red light and promotes seed germination, whereas Pfr is converted into Pr in the presence of far-red light. Pfr translocates to the nucleus, where it controls the transcription of GA- and ABA-related genes by altering the stability or mRNA abundance of several transcription factors [93].

In summary, the germination of a seed is dependent on the precise balance of ABA and GA. ABA is a key player for the entrance and the establishment of seed dormancy and is necessary for the quiescent stage of the seeds, whereas the GA-mediated pathway is an important regulator for the promotion of seed germination in favorable conditions. Many components of the ABA and GA pathways, i.e., ABI3, ABI4, ABI5, RGL2, MFT, and DOG1, effectively control the germination of the seed. Moreover, auxin, jasmonic acid, brassinosteroids, and ethylene modulate the ABA pathway in seeds, which indicates that seed germination is an extensive process which occurs due to the crosstalk of phytohormones.

### 2.3. Optimum Temperature Enables Seed Germination

High or low temperature can cause a delay in the germination of seeds due to the obstruction of various molecular and physiological processes [94]. Time to germinate extends when temperatures remain low [95], and dormancy procrastinates due to prolonged low temperatures experienced by the mother plant before flowering. The temperature of the soil in which the seed is sown also regulates seed germination [8]. For optimal germination, most seeds require temperatures between 15–30 °C. The delay in germination at the low temperature of 5 °C prevents the protrusion of the radicle, which ensures that germination occurs in suitable conditions that lead to the seedling’s successful establishment. Delay in the germination time prolongs the imbibition period of the seed, leading to the production of necessary proteins synthesis in wheat embryos. These proteins play a significant role in breaking the dormancy of the seeds, which is why cold seems to have an advantageous effect on germination [96]. However, the nature of germination and intensity of response varies among plant species and in different cultivars of the same species [97,98]. On the other hand, when heat stress was applied, seeds showed a decrease in metabolites and rate of reserve mobilization, which is directly linked to the loss of seed viability [99]. The efficiency of seed reserve utilization decreases with the increase in temperature [100]. Wheat seeds have the capacity to germinate at 45 °C but the germination percentage remained only 12% due to cell death and embryo damage [101,102]. Under moderate heat stress, the cellular damage may occur if the stress prevailed at a longer period of time [96]. These injuries may include protein denaturation, aggregation, and increased fluidity of membrane lipids. Indirect and slower heat injuries cause inactivation of enzymes in the chloroplast and mitochondria, inhibition of protein synthesis, protein degradation and loss of membrane integrity [103]. High temperature negatively affects seed development due to the accumulation of ROS (especially H_2_O_2_) due to lipid peroxidation. H_2_O_2_ has a dual role; it acts as signal transduction in cell growth and development and also triggers a wide range of stressful environments that lead to programmed cell death (PCD). In response to ROS, many detoxifying enzymes, i.e., SOD, APX, CAT, and GR, are produced and are disrupted and inactivated due to the heat stress [104]. The inability of the cell detoxification mechanism to work against ROS causes the rise of ROS inside the cell that damages internal organelles, especially cell walls, membranes, and mitochondria, which leads to endosperm decay [105]. Temperature, phytohormones, ROS, and light regulate seed germination in four sequential steps, i.e., embryo growth, testa rupture, endosperm rupture (radicle emergence), and growth of epicotyl for seed germination [106].

Low temperature promotes DOG1 transcript accumulation, inducing prolonged seed dormancy due to zygotic tissues rather than seed coat tissues, which are not living. Pericarp thickness and integrity plays a decisive role in seed dormancy, which can be released through seed stratification [107]. The depth of the dormancy depends on the presence of DOG1 proteins in the mature seeds. In addition to the developmental pathway in zygotic tissues, seed coat pigmentation—which is determined by flavonoid concentrations—plays an important role in seed fate. High seed pigmentations cause strong seed dormancy, whereas the seeds with transparent testae have high germination rates. Transcripts of flavonoid biosynthetic pathway genes were upregulated in response to the low temperature, including the regulatory MYB, basic helix–loop–helix (bHLH) transcription factors, and procyanidin-synthesizing enzyme *BANYULS* [107,108]. However, in Arabidopsis thaliana, flavonol-deficient fls1-3 mutant—which shows a severe reduction in flavonol concentration but normal soluble proanthocyanidins (PAs)—indicated that flavnols do not have a significant role in seed germination. In contrast, three alleles of *BANYULS/ANTHOCYANIDIN REDUCTASE* demonstrated low dormancy which had required levels of PA. However, leucanthocyanidin dioxygenase (ldox/tt18) and dihydroflavonol reductase (dfr/tt3-1) mutants which produced less PA showed reduced germination. Furthermore, temperature during seed production affects tannin biosynthetic gene expression because tannins are known to regulate seed permeability. At low temperatures, procyanidin content increases due to the high production of monomers that polymerize to form condensed tannins. Therefore, with the change in temperature, the synthesis of procyanidins is altered, which might be due to the changed transcript abundance regulation of the enzymes required for tannin synthesis [4].

In high temperatures, FUS3 controls the ABA/GA ratio by negatively regulating GA and positively regulating ABA during postembryonic development. Seeds imbibed at 32 °C rapidly degrade FUS3 mRNA stored in the seed and induce de novo FUS3 mRNA synthesis within 12 h, which leads to the accumulation of FUS3 proteins by 48 h due to translational or posttranslational regulation induced by HS. Interestingly, FUS3 can only be detected in the seeds which commence their germination at 32 °C, whereas FUS3 was not detected in those seeds which remained in a dormant state. This indicates that FUS3 is only active in a small developmental window to delay the seed germination process through the activation of de novo ABA biosynthesis. Among the ABA biosynthetic genes, NCED1, NCED5, NCED9, ABA1, and ABA2 showed increased expression at 12 and/or 24 h, while CYP707A2, the most abundant ABA catabolic gene during germination, showed a transient reduction of expression at 12 h. These changes in gene expression are consistent with previous quantifications of transcript levels of several ABA metabolic genes during imbibition at 34 °C [109]. The increase in FUS3 mRNA level at 12 and 24 h parallels that of the ABA biosynthetic genes. Interestingly, all ABA metabolic genes identified in this microarray, with the exception of NCED1, contain RY elements, which interact with B3-domain proteins [110,111]. Since FUS3 positively regulates ABA levels, these genes may be directly regulated by FUS3 and/or other B3-domain proteins [112].

### 2.4. Endosperm Decay, A Prerequisite for Embryo Growth

Germination is deemed successful when the radicle protrudes through the outer covering layers of the seed, leading to seedling establishment (Figure 3) [112]. During seed ripening, protein oxidation by ROS occurs in the dry seeds, which intensifies with the perpetual dry state of the seeds [113,114]. Protection against oxidative damage by ROS is provided by small amounts of antioxidants present in the seed, mainly glutathione (GSH). ROS oxidizes GSH to its dimer GSSG, which accumulates during seed storage [115]. The lipophilic antioxidant tocopherol prevents membrane lipid peroxidation, improving seed longevity and germination [116]. A third antioxidant is ascorbate, which is present in minute amounts to regulate redox reactions in the dry state [117]. A controlled process of oxidation leads to the loss of dormancy, but if the oxidative damage intensifies, deterioration and loss of viability can occur. Dry seeds contain up to 10,000 mRNA transcripts, representing the installed mechanisms for surviving severe dry conditions and the potential oxidative damage from ROS (Table 3) [118]. The accumulation of transcripts in seeds is tissue-specific: transcripts that accumulate in the embryo are different from those accumulated and activated in the endosperm during germination and development [119].

Testa rupture begins when the dry seed starts to imbibe water from the soil [144]. During the imbibition process, the seed rapidly swells and changes in size and shape. The micropyle is the major entry point for water uptake in the Arabidopsis seed, and this phenomenon has also been observed in pea, tobacco, and other species by H-NMR image analysis [114]. The Arabidopsis seed surface is characterized by volcano-shaped cell wall structures called columellae, from which mucilage is released during imbibition [145]. This mucilage is composed of rhamnogalacturonan pectins and cellulose arranged in an outer water-soluble layer, and an inner layer is covalently bound to the testa by cellulose microfibrils [146]. This mucilage helps the seed to travel long distances for effective dispersal by attaching to the skin of animals; it also assists in germination under drought and salt stress conditions [147]. Imbibition causes the leakage of cellular solutes that triggers the germination process and reduces the concentration of inhibitors. Leakage also damages cellular membranes because of rapid and non-uniform rehydration after a long period of dehydration and storage [148]. The seed activates repair mechanisms for membranes and proteins whose aspartyl residues have been damaged by conversion to isoaspartyl. Isoaspartyl methyltransferase can catalyze the transformation of isoaspartyl back to its original form.

The testa ruptures in response to the cumulative effects of many factors, including low pigmentation, hormone sensitivity, and altered morphology (Figure 4) [149]. During the first phase of imbibition, the seed reaches its capacity for water uptake, attains a constant size and shape, and moves into phase Ⅱ, i.e., testa rupture. During phase Ⅱ, the water content stays the same; the duration of this phase varies, and it ends with the rupture of the testa [150,151]. Testa rupture begins at the micropylar end that covers the radicle. Tobacco seeds have predetermined breaking points for testa rupture that are assisted by channel-like structures underlying the ridges. After rupture, phase Ⅲ begins: the endosperm ruptures and the radicle protrudes from the seed, and this is called germination sensu stricto [8].

The development of the endosperm occurs in several phases: formation of the nuclear endosperm, cellularization, differentiation, maturation, and cell death [156]. There are four types of cells in the endosperm: starchy endosperm, the aleurone layer, transfer cells, and the region surrounding the embryo. The starchy endosperm is the main source of nutrients and energy for embryo development, seed germination, and seedling establishment [157]. The endosperm next to the radicle is called the micropylar endosperm (ME). The ME expresses endosperm-specific genes that assist in the loosening of the cell wall [158]. Seed cell wall thickness varies between different plants. Thick cell walls like those of tomato seeds must undergo a long process of weakening to enable seed germination. By contrast, the seed cell walls of Arabidopsis are thin and made up of an aleurone-like cell layer. The aleurone layer functions to inhibit seed germination by acting as a mechanical barrier and also provides nutrients to germinating seeds [159]. The aleurone layer secretes hydrolytic enzymes that catabolize proteins and starch to provide nitrogen and carbon for seed germination [160]. The Arabidopsis endosperm accumulates lipids in the form of triacylglycerols (TAGs) that are catabolized into sucrose by gluconeogenesis during and after seed germination. Lipids stored in the endosperm differ from proteins stored in the embryo in both their chemical composition and their mechanism of catabolism [161]. During germination, cress seeds accumulate proteins that are involved in protein folding, protein stability, energy production, and defense. Their abundance is linked temporally, spatially, and hormonally to the weakening and rupturing of the ME, and the ME of cress therefore has a regulatory function through cell wall modification and does not act solely as a source of nutrients for seed germination [162].

The loosening of the endosperm cell wall is a prerequisite for seed germination. The cell wall is composed of cellulose, hemicellulose, and pectin and functions to regulate seed germination [163]. Cellulose is a linear polymer of β(1→4) linked glucose units, whereas hemicellulose and pectin are heteropolysaccharides composed of different sugar monomers [164]. The activities of cell-wall-remodeling enzymes (CWREs) are important for the loosening of the cell wall. The cell wall reacts to its surroundings, and ABA inhibits germination by preventing the weakening of the endosperm, whereas gibberellin stimulates germination by regulating the abundance of CWREs [165]. Delay of germination 1 (DOG1) is under negative epigenetic regulation mediated by KKYP/SUVH4, LDL1, and LDL2, which positively regulates seed dormancy. DOG1 metabolizes GA in a temperature-dependent manner, which increases the coat dormancy. DOG1 differentially regulates the expression of GA biosynthesis genes such as *GA3ox1* and *GA20ox* at 18 °C and 24 °C, respectively, which leads to the inhibition of genes encoding CWREs: expansin 2 (EXPA2), EXPA9, and xyloglucam endo-transglucosylase 19 (XTH19), but only at 24 °C. Therefore, DOG1 regulates the appropriate time of germination according to the environmental temperature. PCD of aleurone cells during germination enhances the transport of stored nutrients required for seed germination and seedling growth. PCD of the starchy endosperm causes it to degenerate, making way for radicle protrusion [166]. GA induces PCD in the aleurone layer and improves the germination prospects of the seeds. GA-deficient tomatoes are unable to germinate unless they are provided with exogenous GA, although the radicle does emerge and protrude if the endosperm is removed [167]. Therefore, inhibition by ABA can be eliminated if the mechanical barrier of the endosperm is removed [168]. When cell wall loosening has occurred, water can move inward to increase turgor pressure and generate cell expansion. This allows for embryo growth, depending upon the extensibility of the cell wall [163].

In summary, seed dormancy and seed germination respond to an interplay of endogenous and exogenous factors [169].

## 3. Conclusions and Future Perspectives

The germination of the seed is a prerequisite to attain the maximum potential of a crop. It requires a deeper understanding of internal and external factors affecting the seed germination and subsequent emergence so that they can be optimized to achieve the maximum crop production. Phytohormone-, light-, and temperature-activated molecular pathways are intertwined, which excites the ROS production that leads to the germination of the seed. Seed germination therefore requires internal responsiveness to external environmental cues.

Reproduction is an important phase in the plant life history. Therefore, the genes responsible for seed dormancy and seed germination have remained under the strongest selection during the course of evolution. Although it has been well established that plant hormones, along with environmental factors, regulate dormancy–germination transition through a complex network, there are several gaps that are still needed to be filled to increase comprehension of this process.

First, ABA suppresses GA synthesis and is a key dormancy inducer; however, we know very little about GA biogenesis during seed dormancy establishment in seeds, which requires further investigation. Secondly, where is the precise location of ABA and GA synthesis molecular activities are taking place inside the seed. Do ABA and GA synthesize de novo at dormancy and germination sites? Previously, some basic studies were performed by developing seed coat bedding assay to demonstrate that ABA is synthesized de novo in the seed coat in an RGL2-dependent manner and thus represses germination of the embryo. However, this demands further study to identify the precise threshold that ABA needs to achieve to halt seed germination. Subsequently, the actual position of GA biosynthesis must also be identified.

Third, during seed development, the increase in auxin production raises the question: what is the molecular pathway that is responsible for its production? Fourth, ABA and auxin function synergistically to positively regulate seed germination, whereas GA acts antagonistically against auxin and ABA to enhance the prospects of seed germination. However, the detailed molecular mechanisms underlying these synergistic and antagonistic relations are largely undiscovered, including the equilibrium between ABIs, DOG1, DEP1, and SPT (Figure 1) and downstream targets of these transcription factors, which function to improve seed germination prospects.

Fifth, environmental cues, such as light and temperature, play a significant role in maintaining seed dormancy and germination. In this regard, the foremost question is that of how seeds sense temperature changes. There is a need to identify seed temperature sensors using biological approaches combined with biophysical and biochemical techniques. Receptor-like kinsases (RLKs) are key components in triggering plant response according to cold and heat stress, which prompts us to propose that these are involved in the temperature sensing, though there is a need to investigate hypothesis and shed new light on temperature-sensing mechanisms in seeds. Moreover, there is a need to analyze the effects of abrupt fluctuations in temperature on phytohormones, which regulate seed germination and dormancy.

Sixth, changing climatic conditions have greatly challenged the ability of the seeds to germinate over diverse geographic landscapes. Therefore, the identification of qtls responsible for providing plasticity against natural variations may provide useful strategies that ensure that plants can germinate and grow well to adapt to dynamic environments. Another interesting finding is that the endosperm is capable of sensing light signals and interacts with the embryo through bidirectional communication, which provides the evidence that the endosperm is not merely a source of nutrients but that it also controls seed germination and controls embryonic signals through actively secreting signals. Examining the function of individual seed cell types, including the endosperm, will provide an excellent model for understanding the mechanisms of cell-to-cell communication. It will also provide insight into how cell–cell communication directs or coordinates the systemic responses of the seed. Continued study into the role of the endosperm will facilitate the application of seed biology knowledge to the development of robust and sustainable agricultural practices.

## Figures and Tables

**Figure 1 ijms-23-08502-f001:**
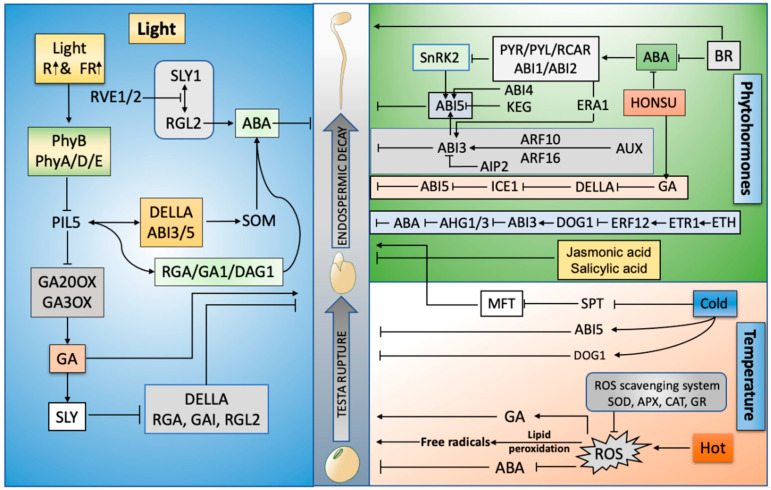
Graphical abstract: driving forces of seed germination: phytohormones, high temperature, light, and endosperm decay. R (Red), FR (Far-red), Sleepy (SLY), ABA (Abscisic acid), GA (Gibberellins), Reactive oxygen species (ROS), Reduced Dormancy 5 (RD05), RGA-LIKE2 (RGL2), SPINDLY (SPY), Teosinte branches 1/cycloidea/proliferating cell factor (TCP), Ethylene (ETH), Indole-3-acetic acid (IAA), Salicylic acid (SA), PHY-INTERACTING FACTORS (PIF), Delay of Germination (DOG), Brassinosteroids (BRs), ABA HYPERSENSITIVE GERMINATION1 (AHG1), Super oxidase dismutase (SOD), Ascorbate peroxidase (APX), Catalase (CAT), Glutathione reductase (GR), MOTHER OF FT and TFL1 (MFT), ABA-insensitive (ABI), Phytochrome (Phy), Serine palmitoyltransferase (SPT), auxin-responsive factors (ARF), Ethylene responsive transcription factor (ERF), ICE1 (Inducer of CBF Expression 1), Diacylglycerol (DAG), Auxin (AUX), Repressor of ga1-3 (RGA), ABI3-interacting protein (AIP), ETHYLENE RESPONSE1 (ETR1), SOMNUS (SOM, a set of 98 genes), Resolvin E1 (RVE1).

**Figure 2 ijms-23-08502-f002:**
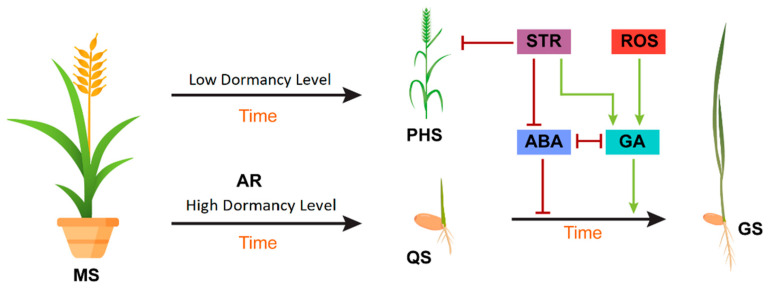
Model showing the effects of extrinsic and intrinsic factors on seed dormancy and germination. During the maturation of the seed (MS), intrinsic ABA is upregulated, and GA is downregulated to inhibit preharvest sprouting (PHS) on the mother plant. After harvest, stratification (STR) and reactive oxygen species (ROS) increase GA biosynthesis and repress ABA biosynthesis, turning the quiescent seed (QS) into a germinating seed (GS). Red bars indicate an inhibition effect, whereas green arrows indicate a promotion effect.

**Figure 3 ijms-23-08502-f003:**
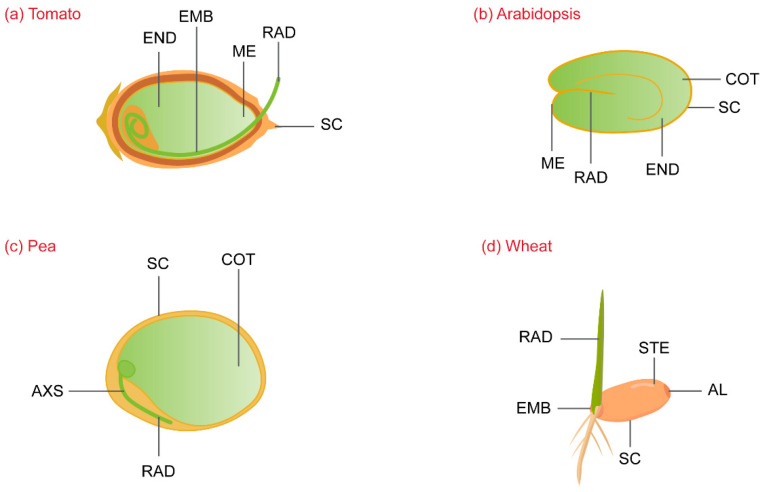
Illustration of the embryo and endosperm in the seeds of tomato (**a**), Arabidopsis (**b**), pea (**c**), and wheat (**d**). Endosperm (END), embryo (EMB), micropylar endosperm (ME), radicle (RAD), seed coat (SC), cotyledons (COT), embryonic axis (AXS), starchy endosperm (STE), aleurone layer (AL).

**Figure 4 ijms-23-08502-f004:**
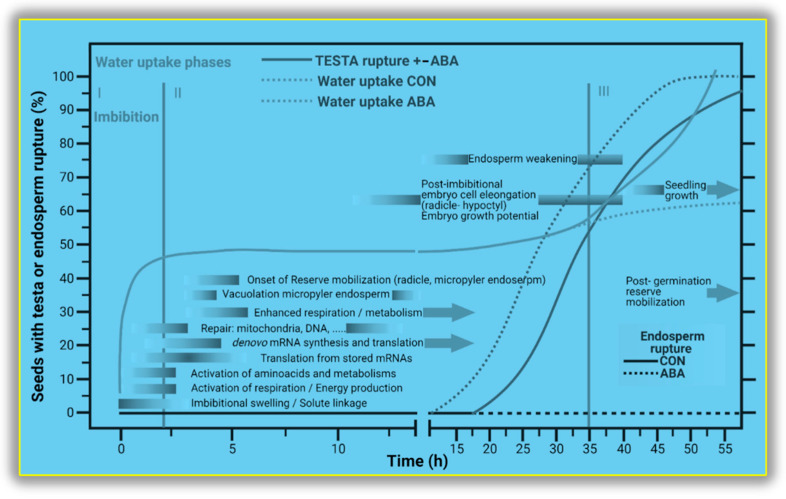
Timeline of crucial processes during the germination of eudicot seeds that exhibit separate testa and endosperm rupture (two-step germination). Water uptake, testa and endosperm rupture, and the effect of ABA on these processes are shown for Arabidopsis thaliana seed; control and without hormone CON. Critical biochemical, biophysical, and cellular events during germination are triggered by water uptake and are shown in the diagram. Water uptake phases: Imbibition (I), Post-imbibition (II), and Post germination reserve mobilization (III). The diagram is based on the understanding from [152,153,154,155].

**Table 1 ijms-23-08502-t001:** Genes involved in seed dormancy and germination.

Name of Gene	Mutant Dormancy Level	General Description of Gene	References
*ABI3*	Decreased	Positively regulates ABA signaling and represses seed germination	[27]
*ABI4*	Decreased	Positively regulates ABA signaling and represses seed germination	[28,29]
*ABI5*	Not Changed	Positively regulates ABA signaling and represses seed germination	[30,31]
*NCED5*	Decreased	ABA-biosynthesis gene; the ABA content is decreased	[32]
*CYP707A1/2*	Enhanced	ABI4 negatively regulates its transcription	[33]
*GAI/2*	Enhanced	GA-biosynthesis genes; GA content is decreased in mutants	[23]
*GA2oxs*	Decreased	GA-inactivate genes; GA content is upregulated in mutants	[34]
*RGL2/SPY*	Enhanced	GA signaling is blocked in mutants	[35]
*MYB96*	Decreased	Decreases transcription of *ABI4* and some ABA biogenesis genes	[36,37]
*DOG1*	Enhanced	ABA sensitivity of *dog*1 seeds is unchanged	[38,39]
*SUVH4/SUVH5*	Enhanced	Repress *DOG1* and *ABI3* transcription	[40]
*LDL1/LDL2*	Enhanced	Repress seed dormancy by negatively regulating DOG1	[17]
*WRKY41*	Decreased	WRKY41 directly promotes *ABI3* transcription	[41]
*RAF10/RAF11*	Decreased	Directly enhances *ABI3* transcription	[42]
*DEP*	Decreased	Promotes *ABI3* transcription	[43]
*SPT*	Decreased in *Ler* but enhanced in *Col* background	Opposite roles in *Ler* and *Col* ecotypes	[44,45]
*ARF10/ARF16*	Decreased	ARF10/ARF16 directly promote *ABI3* transcription	[46]
*BIN2*	Not mentioned	Phosphorylates and stabilizes ABI5 to enhance ABA signaling	[47]
*PKS5*	Not mentioned	Phosphorylates ABI5 (Ser42) and controls transcription of ABA-responsive genes	[48]
*HONSU*	Enhanced	A PP2C protein that impairs ABA signaling	[20]
*RDO5*	Enhanced	Its ABA sensitivity and content remain unchanged	[21]
*ABI1/2*	Decreased	Dominant-negative mutants; the mutated proteins cannot interact with ABA receptors	[49]
*CHO1*	Decreased	Acts upstream on ABI4 genetically	[50]
*OsAP2-39*	Decreased	Promotes OsNCEDI and OsEUI, thereby enhancing ABA biogenesis and impairing GA accumulation	[51]
*DDF1*	Decreased	Directly promotes GAox7 and thus decreases GA content	[52]

**Table 2 ijms-23-08502-t002:** The factors involved in the light regulation of seed dormancy and germination.

Protein	Locus	Possible Biochemical Function	Loss of Function Phenotype	References
PHYA	AT1G09570	Regulates GA/ABA biosynthesis and signaling	Reduced germination in FR and R	[64]
PHYB	AT2G18970	Regulates GA/ABA biosynthesis and signaling	Reduced germination in FR and R, increased dormancy	[65]
PHYC	AT5G35840	Regulates GA biosynthesis	Increased germination in FR	[66]
PHYD	AT4G16250	Regulates GA biosynthesis	Reduced germination in FR	[66]
PHYE	AT4G18130	Regulates GA biosynthesis	Reduced germination in FR	[65]
PIF1	AT2G20180	Directly activates SOM, RGA, and GAI expression; indirectly activates ABA biosynthesis genes and represses an ABA catabolic gene	Increased germination in FR	[67]
SOM	AT1G03790	Regulates the expression of GA and ABA metabolic genes	Increased germination in FR	[67]
JMJ20	AT5G63080	Increases H4R3me2 in GA3ox1 and GA3ox2 chromatin	Reduced germination in jmj20jmj22 double mutant seeds in R	[68]
JMJ22	AT5G06550	Increases H4R3me2 in GA3ox1 and GA3ox2 chromatin	Reduced germination in jmj20jmj22 double mutant seeds in R	[68]
CTG10	AT4G19330	Promotes PIF1 degradation	Reduced germination in FR	[69]
COP1	AT2G32950	Promotes PIF1 degradation	Reduced germination in FR	[70]
SPA1	AT2G46340	Promotes PIF1 degradation	Reduced germination in spaQ, R, and FR	[70]
SPA2	AT4G11110	Promotes PIF1 degradation	Reduced germination in spaQ, R, and FR	[70]
SPA3	AT3G15354	Promotes PIF1 degradation	Reduced germination in spaQ, R, and FR	[70]
SPA4	AT1G53090	Promotes PIF1 degradation	Reduced germination in spaQ, R, and FR	[70]
COP10	AT3G13550	Enhances PIF1 stability	Increased germination in FR	[71]
DET1	AT4G10180	Enhances PIF1 stability	Increased germination in FR	[71]
HEC2	AT3G50330	Blocks PIF1 transcriptional activity	Reduced germination in R	[72]
LUH	AT2G32700	Serves as a co-regulator of PIF1	Increased germination in FR	[73]
HFR1	AT1G02340	Blocks PIF1 transcriptional activity	Reduced germination in FR	[74]
CSN1	AT3G61140	Stimulates RGL2 degradation and further inhibits ABI5 activity	Delayed/ reduced germination	[75]
CSN5A	AT1G22920	Inhibits ABI5 accumulation	Delayed/ reduced germination	[75]
FHY3	AT3G22170	Directly activates ABI5 expression	Increased germination in ABA	[76]
FAR1	AT4G15090	Activates ABI5 expression	Increased germination in ABA	[76]
HY5	AT5G11260	Directly induces ABI5 transcription	Increased germination in ABA	[77]
BBX21	AT1G75540	Interferes with HY5′ binding to ABI5	Reduced germination in ABA	[78]
IMB1	AT3G07610	N/A	Reduced germination in ABA	[79]
CCA1	AT2G46830	Regulates the expression of GA/ ABA related genes	Reduced dormancy in cca1lhy; overexpression increases dormancy	[80]
LHY	AT1G01060	Regulates the expression of GA/ABA related genes	Reduced dormancy in cca1lhy; overexpression increases dormancy	[80]
PIF6	AT3G62090	N/A	Increased dormancy	[81]
RVE1	AT5G17300	Directly inhibits GA3ox2 expression, prevents RGL2 degradation	Reduced dormancy, increased germination in R	[82]
RVE2	AT5G37260	Directly inhibits GA3ox2 expression	Reduced dormancy, increased germination in R	[82]
DAG1	AT3G61850	Directly inhibits GA3ox1 expression	Reduced dormancy and increased germination in R	[83]
SPT	AT4G36930	Induces RGL3 and ABI5 expression in the *Col* background; suppresses RGA and ABI4 expression in the *Ler* background	Reduced dormancy in *Col* background; increased dormancy in *Ler* background	[84]
ELF3	AT2G25930	Inhibits DOG1 expression	Increased dormancy	[85]
LUX	AT3G46640	Directly inhibits DOG1 expression	Increased dormancy	[85]
PKL	AT2G25170	Inhibits DOG1 expression by regulating its H3K27me3; interacts with LUX	Increased dormancy	[85]

**Table 3 ijms-23-08502-t003:** Transcription factors expressed in endosperms involved in the germination of seeds.

Gene	Species	Binding Site	Target Gene	Expression	Phenotypes	References
**bZIP Transcription Factors**						
Opaque2 (O2)	Maize	GCN4-like motif (TGASTCA)	a-zein, 32 kDa albumin (b-32)	Endosperm-specific	Soft and chalky endosperm with high lysine and tryptophan	[120]
BLZ1	Barley		Ltr1	Endosperm, roots, and leaves	NA	[121]
BLZ2	Barley		Hor-2	Endosperm-specific	NA	[122]
SPA	Wheat		LMWG-1D1	Seed-specific	NA	[123]
RISBZ1	Rice		OsLKR/SDH	Endosperm-specific	Higher lysine contents	[124]
TRAB1	Rice	ABRE (ACGT box)	Osem	Embryo roots and leaves	NA	[125]
HvABI5	Barley		HVA1, HVA22	Aleurone layer	HvABI5 RNAi inhibits the ABA activation of ABRC-GUS	[126]
AtABI5	Arabidopsis		AtEm6, AtEm1	Embryo and micropylar endosperm	Reduced sensitivity to ABA inhibition of germination	[127]
AtbZIP44	Arabidopsis	G box (CACGTG)	AtMAN7	Embryo and micropylar endosperm	Delayed germination	[128]
**DOF Transcription Factors**						
PBF	Maize	Prolamin box (TGHAAAG)	γ-ZEIN	Endosperm-specific	NA	[129]
WPBF	Wheat		α-gliadin	Root, cotyledon, leaf, stem, flower, seeds	NA	[130]
HvDOF24/ BPBF	Barley		Hor-2, Al21, Amy2/32b	Endosperm-specific	NA	[131]
HvDOF23/SAD	Barley		Ltr1, Hor-2, and Al21	Starchy endosperm, aleurone cells, nucellar projection, vascular tissues, and immature embryo	NA	[132]
HvDOF19	Barley		Al21	Aleurone layer and embryo	NA	[133]
**GAMYB Transcription Factors**						
HvGAMYB	Barley	G-ARE (T/C) AAC (A/T) AC	Hor-2 and ltr1	Aleurone layer, starchy endosperm	Transient expression of HvGANYB RNAi blocks gibberellin- induced vacuolation in aleurone cells	[134]
OsGAMYB	Rice	GARE (TAACAAA)	RAmy1A	Aleurone cells and anthers	Defects in gibberellin induced gene expression in the endosperm, incomplete heading, sterile panicle	[135]
AtMYB101, AtMYB33, AtMYB65	Arabidopsis	NA	NA	Endosperm, embryo, anthers; MYB101 is endosperm specific	Defects in gibberellin induced vacuolation in germinating endosperm	[127]
**DELLA Proteins**						
SLN1	Barley	NA	NA	NA	Constitutive expression of α-amylase in aleurone layer, slender plants	[136]
SLR1	Rice	NA	NA	NA	Constitutive expression of α-amylase in aleurone layer, slender plants	[137]
RGL2	Arabidopsis	NA	NA	NA	Inability to secrete ABA from the endosperm	[138]
**B3 Domain Transcription Factors**						
Viviparous (VP1)	Maize	RY/SPH motif (CATGCA)	C1, a regulator for anthocyanin biosynthesis	Embryo and aleurone layer	ABA-insensitive seed, reduced accumulation of anthocyanins in kernels, vivipary	[139]
OsVP1	Rice		Osem	Embryo and aleurone layer	NA	[140]
AtABI3	Arabidopsis		SOMNUS (SOM, a set of 98 genes)	Embryo and endosperm	ABA insensitive seed, severe defects in seed maturation, desiccation intolerant seeds	[127]
HvFUS3	Barley		Hor-2 and ltr1	Embryo, endosperm, and aleurone cells	HvFUS3 complements Arabidopsis *fus3* mutants	[141]
**bHLH Transcription Factors**						
AtPIL5	Arabidopsis	G-box (CAGGTG)	SOMNUS (SOM), GAI, RGA	Both embryo and endosperm in germinating seeds	PhyB independent germination, dissected endosperm secretes ABA in light dependent manner	[142]
**WRKY Transcription Factors**						
HvWRKY38	Barley	W-box (TTGACY)	Amy32b	NA	NA	[143]

## Data Availability

Not applicable.

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
