# Peer review of "Underlying Biochemical and Molecular Mechanisms for Seed Germination"

_ijms, 2022, doi:10.3390/ijms23158502_

Round 1

Reviewer 1 Report

This work aims to summarize the previous studies related to the role of environmental factors, phytohormones, and endosperm decay in seed germination. The issue was dealt with in-depth, with satisfactory bibliographic support. However, in my opinion, the work needs to be more comprehensible. Research gaps, conclusions, and future perspectives need to discuss in a separate section. English editing is required; in some sentences, english errors jeopardize comprehension of the text. 

Author Response

Dear Reviewer,

Reviewer 2 Report

The manuscript is ilustrated and written well. It has some editorial errors that need to be double-checked and corrected before acceptance. 

Line 2-3: Underlying Biochemical and Molecular Mechanisms for Seed 

Germination 

Line 12: , and

Line 13: Please double-check it should be “ jasmonic and salicylic acid”?

Line 15-16: Please edit as follows: The incident light wavelength and low and supra-optimal temperature modulate phytohormone signaling pathways that induce the synthesis of ROS

Line 22: Please double-check the journal requirements all keywords should start with a lowercase 

Line 32: phytohormones [3]  Please add space before [3]

Line 37: germination [5].

Line 65: Please double check the journal requierments for the review articles not sure about the “Main Text” term 

Line 67: Please delete it is repeated Line 66

Line 179: germination [36,37].

Line 229, 304, etc: Please be consistence for subheadings 

They should be in italic and each word start with the uppercase 

Please delete the . at the end of the subheadings and titles 

Line 254: has a dual role, it acts as signal 

Line 259: organelles, especially

Line 259: cell walls

Line 260: to endosperm decay 

Line 260-261: emperature, phytohormones, ROS, and light regulate seed germination 

Line 306: (Fig 3) [68].

Line 308: [69,70]

Line 317: (Table 3) [74].

Line 399:  a deeper

Line 402: leads to 

Line 403: Seed germination, therefore, 

Line 411: Figure 2. 3. 4. all should be bold

Suggestion: You may read and add the following articles for breaking dormancy in different crops as it is a review paper:

Zhang, W.; Qu, L.-W.; Zhao, J.; Xue, L.; Dai, H.-P.; Xing, G.-M.; Lei, J.-J. Practical Methods for Breaking Seed Dormancy in a Wild Ornamental Tulip Species Tulipa thianschanica Regel. Agronomy 2020, 10, 1765. https://doi.org/10.3390/agronomy10111765

Huang, W.C.; Maytona, H.S.; Amirkhani, M.; Wang, D.C.; Taylor, A.G. Seed dormancy, germination and fungal infestation of eastern gamagrass seed. Ind. Crop Prod. 2017, 99, 109–116.

Line 448: Please edit the Refrences according to the journal instruction, Numbers are double as well 

Round 2

Reviewer 1 Report

Delete the subtitle “research gaps and future perspectives” and rename section 3 into Conclusions and future perspectives